# Pathological Changes in Early Medieval Horses from Different Archaeological Sites in Poland

**DOI:** 10.3390/ani14030490

**Published:** 2024-02-01

**Authors:** Maciej Janeczek, Daniel Makowiecki, Aleksandra Rozwadowska, Wojciech Chudziak, Edyta Pasicka

**Affiliations:** 1Department of Biostructure and Animal Physiology, Wrocław University of Environmental and Life Sciences, Kożuchowska 1, 51-631 Wrocław, Poland; maciej.janeczek@upwr.edu.pl (M.J.); aleksandra.rozwadowska@upwr.edu.pl (A.R.); 2Institute of Archaeology, Nicolaus Copernicus University in Toruń, Szosa Bydgoska 44/48, 87-100 Toruń, Poland; makdan@umk.pl (D.M.); wojchud@umk.pl (W.C.)

**Keywords:** paleopathology, horse, Middle Ages, archaeozoology, Poland

## Abstract

**Simple Summary:**

The work is the first comprehensive analysis of equine pathological changes from the Polish territory. The research material was collected from 20 archaeological sites, mainly early medieval settlements, such as strongholds, settlements, towns and horse graves. In the material examined, 186 cases of lesions were found. Of these, 26.9% were lesions of the spine, 39.8% lesions of the limb skeleton and 31.7% lesions of the head including dental pathologies.

**Abstract:**

The work is the first comprehensive analysis of equine pathological changes from the Polish territory. The research material was collected from 20 archaeological sites, mainly early medieval settlements, such as strongholds, settlements, towns and horse graves. In the material examined, 186 cases of lesions were found. Of these, 26.9% were lesions of the spine, 39.8% lesions of the limb skeleton and 31.7% lesions of the head including dental pathologies. Most of the lesions in the limbs involved their distal segments. The vast majority of pathological cases can be linked to animal use. It was found that horses in which pathological lesions were observed were used under cover. In one case, the observed cranial trauma was the cause of death associated with injury to the nasal auricles and large vessels and consequent blood loss and possible shock. It was found that, in some of the cases, the horses started to be used early which affected their organs of motion and spine.

## 1. Introduction

The domestication of the wild horse in the first half of the fourth millennium on the steppes of southeastern Europe was a watershed event, first for pastoralists of the steppes stretching from the Ukraine to the Urals and then for peoples of other regions of Europe, Asia and North Africa [1,2,3,4,5,6,7,8]. It brought about changes in the tactics of warfare and battle, led to the invention of horse-drawn fighting vehicles and contributed to the growing importance of wheeled transport. The functions assigned to this animal by man resulted in the organism, especially its muscles and skeleton, having to assume unnatural positions during movement. This resulted in injuries in various parts of the locomotor apparatus due to trauma and chronic inflammation. This is evidenced by quite a number of works on lesions in horses used in Europe both in prehistoric times, in the Middle Ages and in modern times [9,10,11,12,13]. Pathologies, which have generally been described in these animals since domestication, were most often the result of different uses, including as mounts and for pulling chariots and wagons [13,14,15,16,17].

In the case of the West Slavs, who inhabited the area of modern Poland, among others, in the early Middle Ages, archaeological data and chronicle records indicate that the horse was of military importance. Such use must have caused certain lesions in the skeletal system and in the dentition. However, in the literature on medieval horses used by Slavs living in present-day Poland, there are a few works dealing with pathological changes. They are dominated by descriptions of individual cases or cases of pathology mentioned only on the occasion of analyses of bone material from individual archaeological sites [12,18,19,20,21,22]. Publications attempting to describe pathologies considering several sites are rare [23]. For these reasons, the aim of this article is to add to the current state of knowledge. This was made possible by the authors’ collection of a relatively large number of cases with lesions during a project on the history of the horse in the early days of the Polish state. This paper is a description of the pathology of equine skeletal remains from various archaeological sites with an attempt to interpret it in a sociocultural context.

### The Horse for the West Slavs

The first information about the probable use of horses by the Slavs was provided by Procopius of Caesarea, the principal Roman historian of the sixth century. In his chronicle, Opus magnum (History of Wars), Book V (Proc. Bell. V 27.1-2), he records that among the horsemen of Martinus and Valerian were Slavic Sclaveni and Antes living on the Danube [24,25]. The riding of horses by elite South Slavs in the sixth century is also reported in Theophylact Simocatta, the secretary of Emperor Maurice from 582 to 602 [26] (p. 76). Customs regulations from the early tenth century (903–906 AD, Tariff from Raffelstetten) concerning the Eastern March (present-day Austria) ordered Slavs from Bohemia and Rus to pay a levy on the horse, stallion and mare trade [27] (p. 117). Ibrahim ibn Yaqub, a Jewish traveller, in notes made in approximately 966, reported on the abundance of horses and horse trading by Slavs from the country of Nakon (ruler of the Obotrites) [27] (p. 147). These records clearly suggest not only that the Slavs used horses but also that the animals they bred were distinguished by their high-performance qualities and probably by their appearance. Still other historical records show that Mieszko I, the founder of the Polish state, as one of the countries of the West Slavs, equipped his armed men with horses [27] (p. 148), [28]. Horsemanship played a decisive role in war expeditions and battles fought [27] (p. 155), [29]. Horse meat was consumed as a valued meat, and mare’s milk was also drunk in the form of kumys. Horse hides were used to make leather goods [30], and skates and sled runners were made from radial bones and metapodials [31]. Due to the cost of buying and maintaining a horse, owning and riding one was a privilege and thus an important marker of belonging to the elite of the time. Both historical data and zooarchaeological research point to the important significance of the horse in the religion of the Slavs. This is evidenced by complete and partially associated bone groups [32,33,34,35]. The present paper is a description of the pathology of equine bone remains from various archaeological sites with an attempt to interpret them in a sociocultural context.

## 2. Materials and Methods

The research material was collected from 20 archaeological sites, mainly early medieval sites (7th–13th c.), such as strongholds, settlements, towns and horse graves (Figure 1; Appendix A). Only a few samples have been dated to late medieval times (14th c.). In the early medieval period, the strongholds were elite residences mainly functioning as economic, administrative, religious and political centres. At the creation of the state of Poland, the most important of the strongholds were Dąbrówka, Ostrów Lednicki, Kruszwica, Kałdus, Gdańsk and Tum. In Dziekanowice, Górzyca and Jordanowo were uncovered horse skeletons buried in human cemeteries. On the other hand, in Żółte (site 33), a horse skeleton was discovered in an underwater context near the island as a local centre of trade and religious events. In that research, the macroscopic observational method was used. To carry out a detailed analysis of specific cases, X-rays were taken using a radiographic system, the Fire Flash CR 70 scanner. The X-rays obtained were interpreted using QuantorVet+ version 2.0 software. For the majority of specimens, photographs were taken showing noted modifications of the original morphology of anatomical elements. The Nomina Anatomica Veterinaria [36] was used for the names of the anatomical structures described. The classification proposed by Bendrey [37], among others, was used in the study. The methodology of Loch and Melvin [38] was employed to determine the age. The results obtained were discussed based on contemporary knowledge in veterinary medicine [39,40,41] as well as palaeopathology [42,43].

## 3. Results

Among the materials examined, 186 lesions were found. Of these, 26.9% were spinal lesions, 39.8% were lesions of the limb skeleton and 31.7% were lesions of the head, including dental pathologies, which will be described in a separate article.

### 3.1. Spine

Within the spine, 49 lesions were identified of which the vast majority were found in the thoracic segment (n = 30) and half as many in the lumbar segment (n = 16). Pathological modifications in the other spinal segments can be described as incidental—cervical segment: two cases and caudal segment: one. In the thoracic spine segment, ankylosis of the intervertebral joints was diagnosed and noted in five horses. Along with this disease, calcification of the intervertebral ligaments (*ligg. interspinalia*) was noted within the spinous processes in three cases, being an advanced form of “kissing spine” syndrome. Another lesion identified was advanced spondylosis occurring with the calcification of the longitudinal ligament of the good spine (*lig. longitudinale ventrale*). This condition was found in five horses. Developing lesions were also observed on the mamillary processes of the thoracic vertebrae (*procc. mamillares*) and on the spinous processes (*procc. spinosi*). These led to the anastomosis of the arches (*arcus vertebrae*) and vertebral bodies (*corpus vertebrae*) through the calcification of the intervertebral spaces (i.e., thus, the interlaminar ligaments and intervertebral discs) (n = 3). In the lumbar spine, ankylosis of the intertransverse joints (*artt. intertransversaria*) associated with the calcification of the intertransverse ligaments (*ligg. intertransversaria*) was identified in four cases. Spondylosis was observed in two lumbar vertebrae (on the ventral surface). In addition, generative lesions were found in the cranial and caudal articular processes. The same type of lesion was noted on the vertebral arches and on the spinous process in two cases. Pathologies within the cervical vertebrae were observed on the dorsal hatch edge of the first cervical vertebra (*atlas*). These were characteristic osteophytic creations at the attachment sites of the short head muscles (enthesophytes). They were most likely formed as a result of chronic inflammation (Figure 2). The same vertebra also shows an anomaly in the morphology of the notch of the arch, which has a strongly asymmetrical shape.

### 3.2. Ribs

Three rib lesions were noted. The first was a specimen with signs of healing, which was an immune system response to the inflammatory process caused by the bone fracture. Macroscopically, the long axis lesion, the rib, the healed break line and the large superstructure of new bone as a periosteal reaction were clearly visible (Figure 3 and Figure 4). The described image is characteristic of a posttraumatic lesion. In the second case, the bony reaction on the rib was the result of a nonspecific inflammatory process with the formation of clear fistulas (Figure 5 and Figure 6). There was a purulent process here, probably caused by a bacterial infection. No trauma marks were noted on the specimen examined. The rib came from a horse skeleton discovered in Gdańsk in the box of a rampart surrounding an early medieval castle. Lesions with similar features were also observed in the skull, vertebral column and limbs of this animal [12]. A third case of disease on this skeletal element was found in the same horse. The rib was slightly remodelled as a result of trauma, which, however, did not cause a break in bone continuity.

### 3.3. Limb Skeleton

In the limb skeleton, the greatest number of lesions was found in the distal segment (excluding the toe), i.e., on the carpal, tarsal, metacarpal and metatarsal bones (Figure 7). In the proximal segment of the thoracic limb, relatively few lesions were observed. These included bone lysis on the caudal surface of the medial condyle of the humerus, i.e., at the proximal attachments of the superficial finger flexor muscle (*m. flexor digitorum superficialis*) and the humeral head of the deep finger flexor muscle (*m. flexor digitorum profundus caput humerale*). An enthesophyte was also observed at the site of the tuberosity of the shoulder (*tuberositas deltoidea*), which is the terminal attachment of the shoulder muscle (*m. deltoideus*). The key to this lesion was that, in the horse, the tendon of the deltoid muscle is anastomosed to the infraspinatus muscle, forming a functional unit with it. Another case involved a non-specific generative lesion noted on the radius bone (*radius*) in a horse from Gdańsk (ul. Sukiennicza). Multiple lesions with similar features were observed in the same individual [12]. Within the proximal segment of the pelvic limb, non-specific manufacturing lesions of the pelvis (*pelvis*) were diagnosed on its iliac wings (*alae ossis pubis*) and ischiadic tuber (*tuber ischiadicum*). These lesions should be considered in conjunction with similar lesions on the skull, spine and ribs of the same animal. Similarly, a generative lesion on the distal femur was also found in the same horse [12]. Lesions on femurs from other animals were also reported. These were osteophytes on the femoral neck (*collum ossis femoris*) and on the medial lip of this bone (*labium mediale*). The medial lip is one of the attachments of the gastrocnemius muscle (*m. gastrocnemius*), which co-forms the common calcaneal tendon (*tendo calcaneus communis*) and is therefore an essential stabilising element of the tarsus. In the case of the tibia, enthesophytes were observed at the attachment site of the popliteus muscle (*m. popliteus*) and the deep toe flexor muscle (*m. flexor digitalis profundus*). Such a lesion was recognised in one bone. In the next tibia, an enthesophyte was observed on the medial condyle (*condylus medialis*), i.e., the attachment site of the semitendinosus muscle (*m. semitendinosus*). This muscle affects the hip and knee joints.

Lesions located in the distal limb were among the most numerous observed in the specimens examined. Among these, 15 cases of fusion of the splint metapodial bones (*metapodials*) were particularly numerous, with myelomeningocele without ankylosis of the joint being the next most frequent (Figure 7). Among the 17 bone spavin disorders found, only 4 had ankylosis with metapodial bones. In one case, a very rare ankylosis of the ankle and heel bones with the other tarsal bones was observed (Figure 8). The other lesions involved the attachment sites of the interosseus muscle (*m. interosseus*) and possibly the lateral and medial collateral ligaments.

Twenty-four lesions were diagnosed in the finger. They represent 38.5% of the cases recorded in the limb. There were 10 on the *phalanx proximalis* and 7 each on the *phalanx media* and *phalanx distalis*. In the *phalanx distalis*, generative lesions were observed on the axial and abaxial sides. In one case, they occurred on the ligamentous triangle area of the *phalanx proximalis* (Figure 9).

Three cases of high ringing, or osteoarthrosis of the coronoid joint, were diagnosed in the *phalanx media*. The ossification of the hoof cartilage (*cartilago ungulae*) was found in two *phalanges distales*.

In one *phalanx distalis*, lesions were observed on its flexor surface at the terminal attachment of the deep flexor muscle of the fingers (*m. flexor digitorum profundus*). These are bony outgrowths on the flexor surface of the *phalanx distalis*, which result from damage to the tendon of the superficial flexor muscle of the fingers or to the distal uncinate ligament. In one *phalanx media*, a subchondral cyst was observed in the proximal aspect (Figure 10). In addition, manufacturing changes were observed as a consequence of a chronic inflammatory process (Figure 11).

### 3.4. Head

Within the head, 59 lesions were identified, of which 16 involved skull bones and the remaining teeth. In one case (Ostrów Lednicki, site: 2, inv. no.: I/87/13/99), a cavity in the skull caused by a large diameter sharp tool was identified (Figure 12). The hole was located on the left frontal bone, just above the nasal bone. The injury penetrated deep into the skull, destroying the auricles and part of the situs bone. If kicked, the situs bone would not have perforated, and only the frontal bone would have been damaged and would probably have been crushed. This wound appears to have been the cause of death, as injury to the situs bone causes extremely heavy bleeding. Horses were not slaughtered in such a brutal manner, and therefore, the animal suffered such a wound during a battle episode. The tool with which the wound was inflicted could have been, for example, a battle hammer [22].

In one skull of a horse from Gdansk, non-specific manufacturing lesions were observed on the nasal, left frontal and left maxillary bones in the region of the infraorbital aperture (Figure 13a,b). Similar lesions were located in this skull in the area of the left pterygoid fossa having, through the infraorbital canal and maxillary aperture, a direct connection with the maxillary region, where such lesions were also observed. As similar generative lesions were also observed in this animal on the ribs, spine and limbs, it seems that this was a generalised bacterial infection [12].

The other cranial bone lesions observed were minor manufacturing lesions in the region of the attachment of the nuchal ligament and lesions associated with the inflammation of the maxillary cusps bilaterally in each case.

## 4. Discussion

The recorded statistical distribution of changes in the different parts of the spine can be explained by the way in which horses were used and the course of the biomechanics of the movements dependent on the performance of specific vital utility functions. When horses were used as mounts, the areas most susceptible to injury were the saddle areas [20,43,44,45].

The ankylosis of the intervertebral joints and ‘kissing spine’ syndrome are considered to be an effect of the use of the horse under the mount and may be a consequence of the excessively early allocation to this function or excessively intensive use [20,44]. The lesions lead to a significant reduction in the mobility of this spinal segment and cause a decrease in the functionality of the spine as a whole [46,47].

Spondylosis is not directly related to the animal’s use and shows a correlation with age and genetic predisposition. The manufacturing lesions appearing on the spinous processes and the anastomosis of the thoracic vertebrae are caused by the inflammation of the spinal fixation muscles and interlaminar ligaments. The ankylosis of the lumbar vertebrae is also a characteristic effect found in ridden mounts; however, the condition can also occur in draft horses, i.e., those harnessed to a cart as well as a plough [13,44].

In summary, the observed spinal lesions allow us to conclude that the horses were used for riding. The cause of the lesions was often repeated microinjuries causing the inflammation of the spinal fixator muscles, spinal joints and spinal ligaments, particularly the short ligaments, i.e., the interlaminar, intercostal and intertransverse ligaments. The lesions were undoubtedly chronic in nature. If, due to painful symptoms and the limited capacity of the animal, the load in the activity performed was changed, it was resumed after a short time. Consequently, the previous inflammation returned, leading to the calcification of the soft tissues. Such changes limited the mobility of the spine and caused pain symptoms in the animal, with varying degrees of severity and manifestation. The aetiologies of these changes included the excessively early initiation of the use of horses, the excessively intensive use of horses, the excessive weight of the rider in relation to the horse and ill-fitting saddles. The clinical signs of the observed changes in modern horses are initially lameness and, in the advanced stages, intolerance of the rider and thus loss of ability to be ridden. Even if such a horse allows itself to be saddled, it will still refuse to perform certain manoeuvres, e.g., jumping, which can be dangerous for the rider.

Rib fractures in horses most commonly occur as a result of being kicked by another horse and less commonly as a result of falls [48,49]. Most often, if there is no perforation of the thoracic cavity associated with complications such as lung injury, pericardial sac, pneumothorax and haemothorax, such fractures heal spontaneously with a large periosteal reaction due to respiratory movements [47].

From studies on the frequency of limb bone pathology, the distribution observed in our material is expected for the remains of performance horses [50]. In an animal with an enthesophyte, there appeared to be a limitation in the freedom of movement of the shoulder joint and, consequently, the mobility of the thoracic limb. For this reason, the use of such a horse was hampered. Summarising the limb lesions, it can be concluded that they were all the result of inflammatory processes in the locomotor organ. Such lesions arose in horses that were used too intensively or over a long period of their life, with repetitive movements that are unfavourable to the natural motility of the horse’s body. The exceptions were the lesions in the horse from Gdansk, which was a laminitis victim and in which the lesions appeared to be related to a nonspecific inflammatory process, which can be interpreted as the result of a generalised bacterial infection [12].

The ossification of the splint bones from the metapodials is a result of ossification of the interosseous ligament (*ligamentum interosseum*) located between the mentioned elements [37]. Ossification takes place to varying degrees and can occur completely along the entire bone, as was the case for the individual in Figure 8 The biomechanical processes of this condition are not fully understood. The conversion of the ligament to osteochondrosis at this site is essentially the result of the inflammation of the interosseous ligament (rather than interosseous muscle, which is also in this area) and changes to the cortical bone in its barrel structure with or without the formation of bone spurs. When inflammation is in the active phase, the animal may present with lameness. Lesions may not be palpable. Most commonly, inflammation affects the ligamentum interarticularis (*ligamentum interarticularis*) between the second and third metacarpal bones or between the fourth and third metatarsal bones in the proximal third. The inflammation of the ligamentous process occurs most often in young horses starting regular work. It can also occur in older horses as a result of overtraining. An individual who develops the ossification of the ligament may have lameness that worsens during exercise. When the process is active and therefore visible to the animal owner, use should be stopped for six weeks. This usually results in the lameness abating, as after this time, the new bone tissue is quiescent and inactive.

In addition, horses with divergent limb postures are particularly prone to the ossification of the ligamentous joint between metacarpal bones II and III. Additional bone tissue then appears on the medial side of metacarpal bone II, which is known as Delpech–Wolff’s law. Ossification can also occur spontaneously in older horses—in both cases usually without accompanying lameness. The lesion can also be secondary to trauma to the splint bones, with or without fracture, and can progress without inflammation of the ligamentous joint. According to the classification proposed by Bendrey [51], all cases recorded in the material examined can be classified as grade II.

Bone spavin is an osteoarthrosis of the distal intertarsal joint and/or the tarsometatarsal joint, which is less commonly the proximal intertarsal joint. It occurs during various uses of the horse, i.e., in jumping horses, racehorses, recreational horses and Western horses. The cause of myelomeningocele has long been known [39,40,52]. The onset of the condition is explained by the repeated compression and rotation of the tarsal bones and severe tension in the intertarsal ligament attachments on the dorsal side of the joint. These occur when the animal jumps or when it stops. The consequence of such activities may be the incomplete ossification of the central tarsal bone or the third tarsal bone. A predisposition to bone spavin occurs with an abnormal posture, particularly when the angle of the tarsal joint is less than 150–153° from the tarsus towards the distal pelvic limb subaxial posture—with increased pressure on the tarsal joint. Bone spavin also occurs when a cow stance is present. In this case, the limb at the level of the tarsal joint is directed medially (‘in-at-the-hock’). Another posture in which the joint of the limb can appear bone spavin is over-erect [39,40,52]. Bone spavin can occur in horses that are gently used, although intensive use accelerates and worsens the disease. The sign of the condition is lameness, but this also disappears if ankylosis occurs.

With regard to the three cases of high toe rings found, both low and high toe rings are formed with the formation of bone formations on the bones that make up the joint within the articular ligaments and joint capsule attachments. This process can occur with or without changes to the articular surfaces. There is no breed predilection. It is most common in older horses and is more common in the thoracic limbs than in the pelvic limbs. The medial side of the joint is also more commonly affected. Horses that have to move and stop suddenly and turn and twist during use are more prone. Currently, horses performing such movements are those used for jumping, dressage and Western [39,40,52]. There are several reasons for the appearance of ringing [39,40,52]. Among these, the constant overloading of the joint and repetitive microinjuries to the joint and surrounding structures are mentioned. Further factors are postural defects, such as the convergent/divergent, tight/divergent, steep pastern and consequent overloading of the periarticular structures, incompatibility and instability of the joint. Ringing can be caused by one severe injury or by several less severe but repeated injuries. It can also be caused by osteochondrosis, unrecognised fractures, sprains or perforations of the joint, all of which are secondary to other joint disorders.

Ossifications of the hoof cartilage in foals affect the adaptation of the developing locomotor organ to environmental conditions and the type of movement performed. In principle, the calcification of these cartilages is not considered to be a pathological change unless it is accompanied by the inflammation of the fixation apparatus of the toe. This phenomenon is thought to occur in all horses after the age of three years, with the rate of calcification being highly dependent on genetic factors [41]. A low ring or osteoarthrosis of the hoof joint was found in one *phalanx distalis*.

The ossification of the hoof cartilages is the result of their mineralisation. In the normal state, the flexor surface (*facies flexoria*) of the *phalanx distalis* is smooth and slightly concave. Injury to the tendon of the deep flexor muscle of the toes (*tendo musculi flexoris digitorum profundus*) or, less commonly, to the ligamentum sesamoideum distale impar (*ligamentum sesamoideum distale impar*) within the attachment on the flexor surface can lead to mineralisation and the formation of enthesophytes, which is usually accompanied by the lameness of the animal. Damage to both structures can be associated with patellar syndrome. Arbor syndrome is a broad set of pathologies that can occur in various combinations in the structures that make up the suspension apparatus of the hoof’s arches, in the arch itself (*os seasamoideum distale*) and/or in the synovial bursa of the sole (*bursa podotrochlearis*). Such lesions may be indicative of the so-called arbour syndrome [39,40,52]. However, due to the absence of the rebleeds themselves in our material, this aspect cannot be clearly determined.

## 5. Conclusions

In summary, it can be concluded, with certainty, that the mode of use for riding was predominant among the horses in which pathological changes occurred. The observed changes in the skeletal system were caused by stresses on the musculoskeletal system. The animals were therefore used intensively, it seems, at a relatively young age. A number of changes, such as the vertebral spondylosis and ossification of the hoof cartilage, are caused by the increasing age of the animals and are not the result of their use. Analyses of the pathology noted in the skulls are direct evidence of the topical use of these animals. The case of a horse from Gdańsk (Sukiennicza St.) has already been described [12]. Apart from two lesions, i.e., a healed rib fracture and a cavity in the skull being the probable cause of death, no evidence was observed that could suggest an aggressive relationship between humans and horses. The rib fracture appears to be the result of a fall or a kick and not of human actions. Nonetheless, it is certain that the animals were exploited long and intensively. The changes observed suggest that the animals were used, as is still the case today, too early. In such specimens, the skeletal system is unable to cope with the stresses and strains that arise from performing the unnatural motor functions given to the animal by man. Some lesions, e.g., myelomeningocele with ankylosis, were in a phase when a period of painful symptoms is followed by an asymptomatic period that does not glaringly affect the horse’s performance. It can be presumed that such individuals were still in use. During the period when lameness was present during myelomeningocele, such animals were thus not fit for work and were not culled. In all likelihood, the symptoms of myelomeningocele were sufficiently known to the horse breeders and users of the time that they did not slaughter the animals despite visible lameness and signs of pain. They knew that the symptoms would cease after a certain period of time and that the functional capacity of the individual in question would return. Nor can it be ruled out that some therapeutic treatment was performed and that it was to these that the improvement in the horse’s condition was attributed. Such practices were still known in the nineteenth and twentieth centuries in the folk tradition in the present-day lands of northeastern Poland. One way of treating lameness is by means of poultices made from water, vinegar, whey or human urine. Lameness was also treated by tying a hemp rope or copper wire around the pastern [53].

## Figures and Tables

**Figure 1 animals-14-00490-f001:**
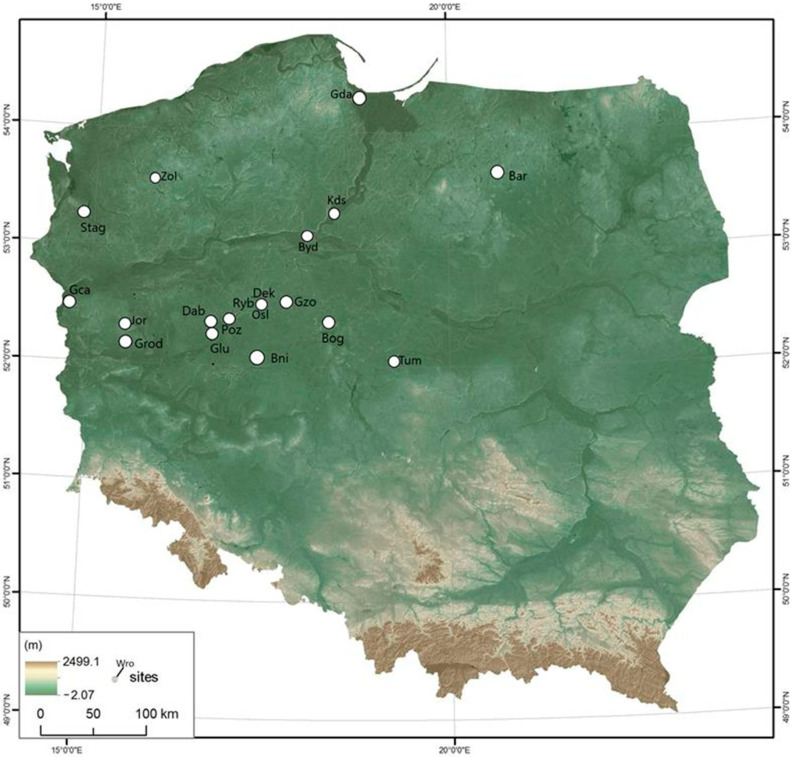
The research materials collected from 20 archaeological sites, mainly early medieval settlements, such as strongholds, settlements, towns and horse graves: Bar—Barczewko; Bni—Bnin; Bog—Boguszyce; Byd—Bydgoszcz; Byd—Bydgoszcz; Dab—Dąbrówka; Dek—Dziekanowice; Gnz—Gniezno; Gro—Grodziszcze Gca—Górzyca; Gda—Gdańsk; Glu—Głuchowo; Jor—Jordanowo; Kds—Kałdus; Poz—Poznań; Osl—Ostrów Lednicki, Ryb—Rybitwy; Stag—Stargard; Tum—Tum (Łęczyca); Zol—Żółte.

**Figure 2 animals-14-00490-f002:**
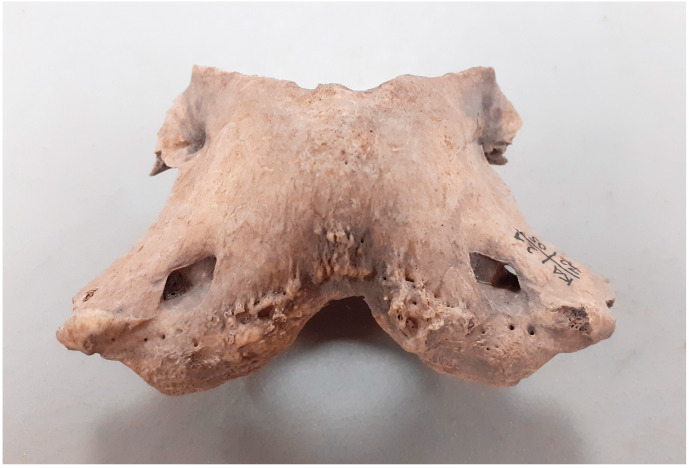
Kałdus, site 2. Enthesophytes on the dorsal surface of the wings and the dorsal arch of the atlas (2nd half 12th–1st half 13th c.).

**Figure 3 animals-14-00490-f003:**
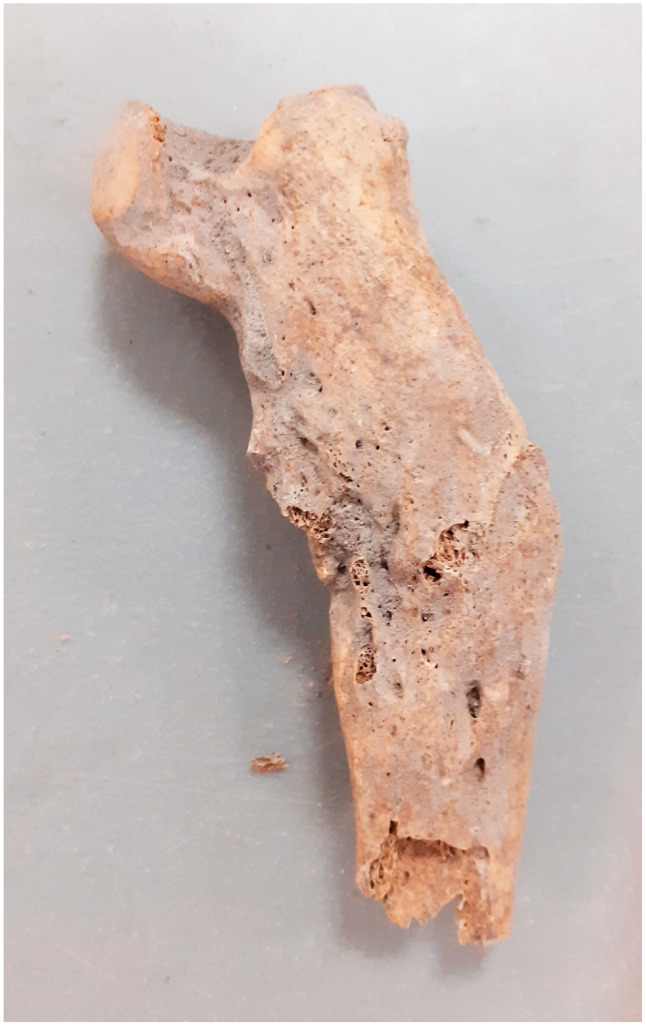
Kałdus, site 3. Healed rib fracture (10th–12th c.).

**Figure 4 animals-14-00490-f004:**
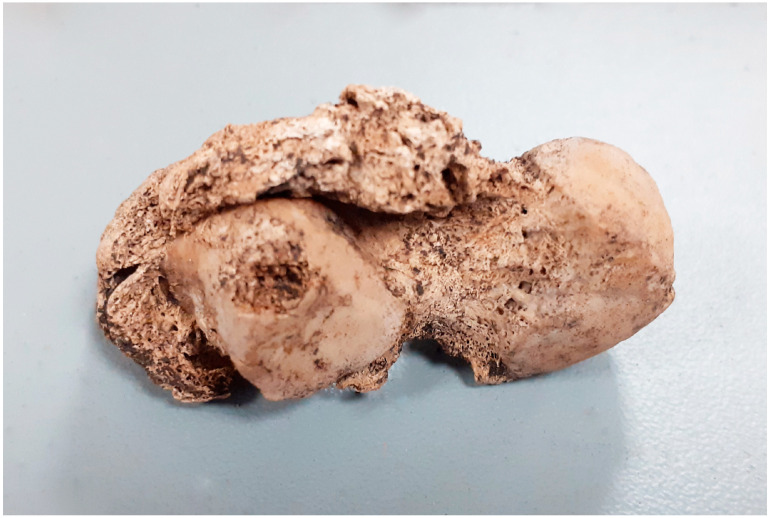
Gdańsk, site 1 (Sukiennicza St.). New bone in the rib neck and rib shaft area. Condition after fracture (11th–mid 12th c.).

**Figure 5 animals-14-00490-f005:**
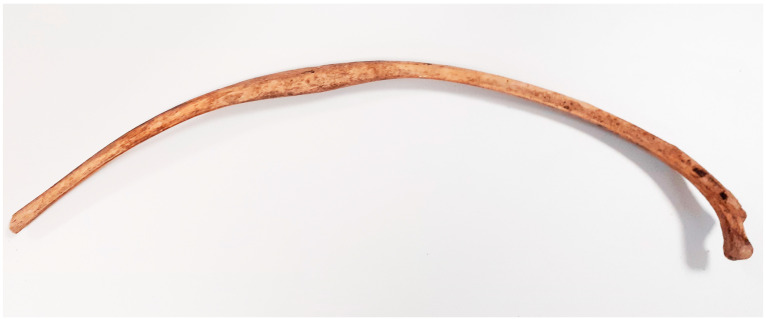
Gdańsk, site 1 (Sukiennicza St.). Post-infectious lesions with visible fistulas (11th–mid 12th c.).

**Figure 6 animals-14-00490-f006:**
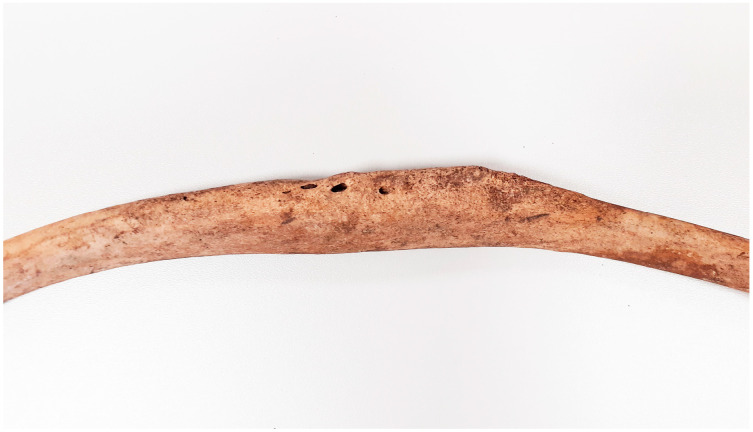
Gdańsk, site 1 (Sukiennicza St.). Post-infectious lesions with visible fistulas (11th–mid 12th c.).

**Figure 7 animals-14-00490-f007:**
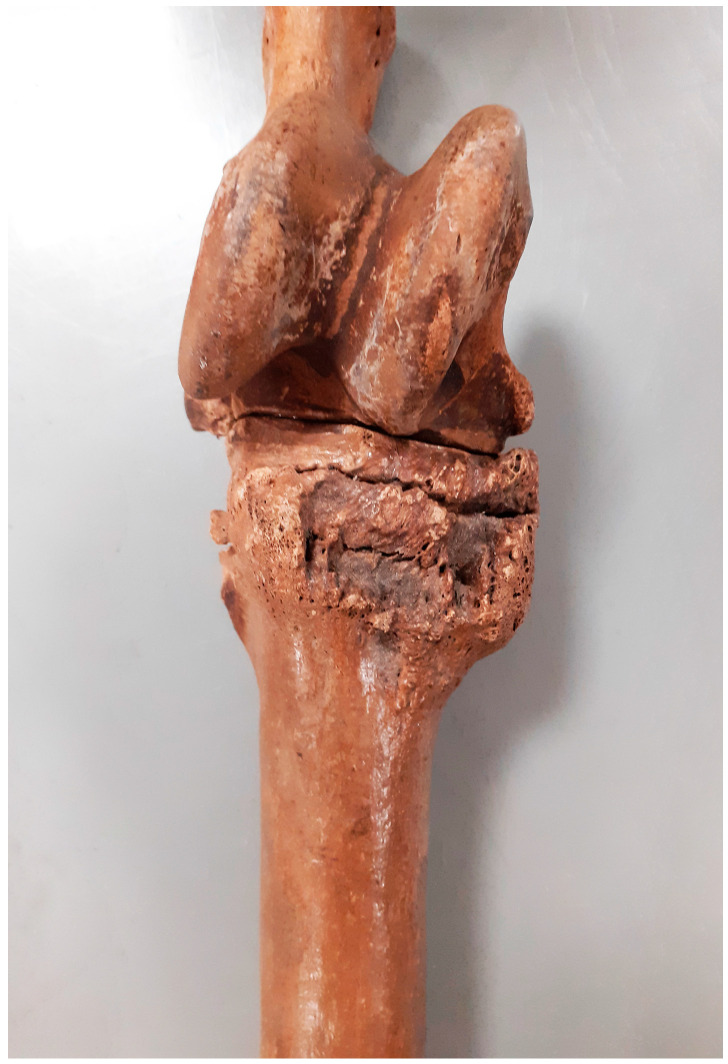
Żółte, site 33. Advanced bone spavin—the skeleton of a male, 8–9 old (late 10th–late 12th c.).

**Figure 8 animals-14-00490-f008:**
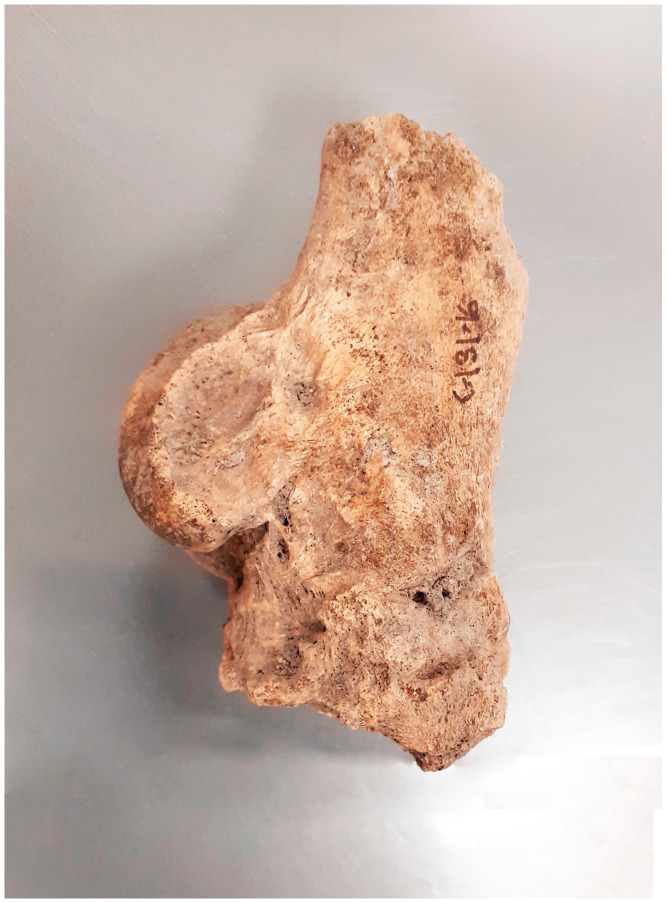
Grodziszcze, site 1. Post-inflammatory fusion of the ankle and heel bones with other tarsal bones observed (1st half 10th–11th/12th c.).

**Figure 9 animals-14-00490-f009:**
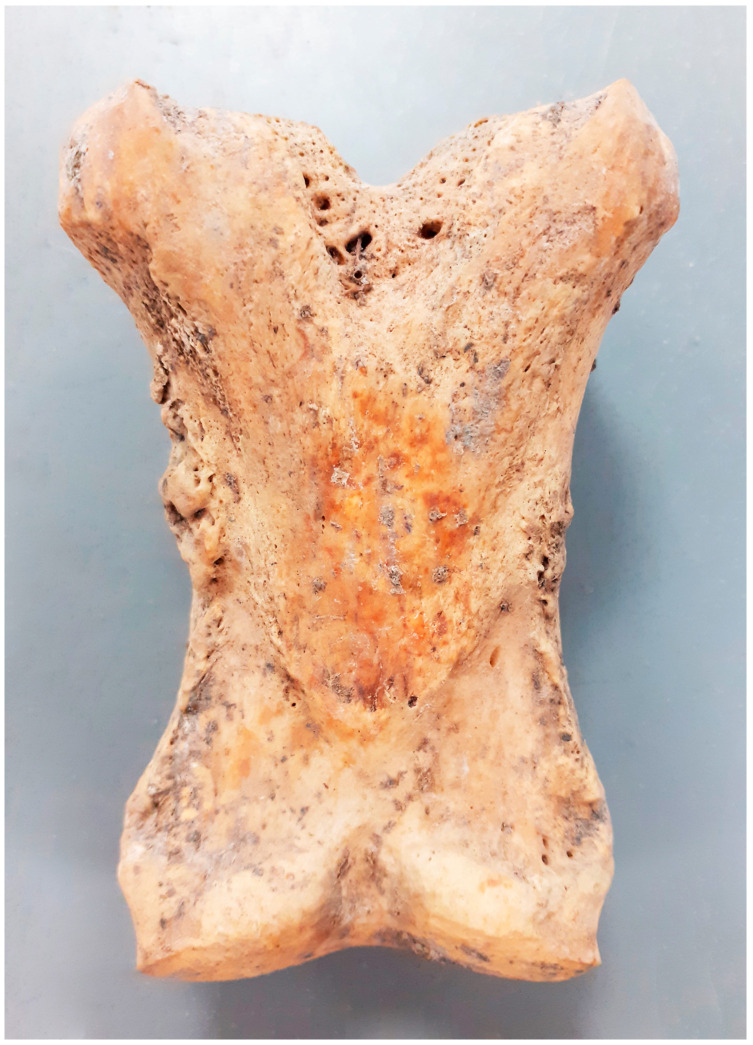
Ostrów Lednicki, site 1. Osteophytes in the area of the ligamentous triangle of the *phalanx proximalis* (12th c.).

**Figure 10 animals-14-00490-f010:**
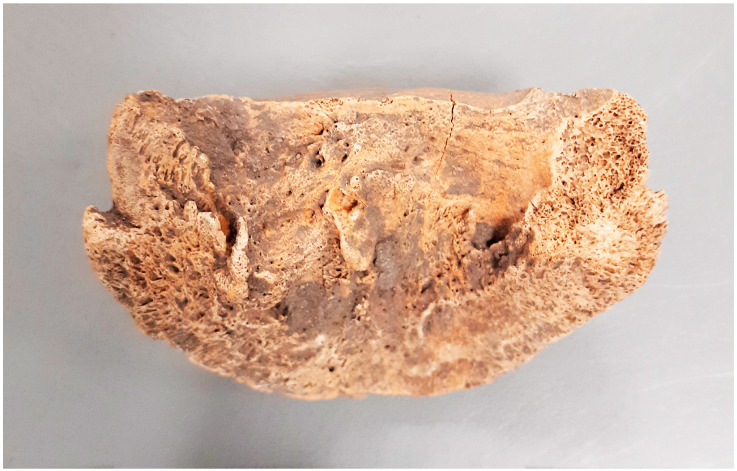
Gniezno, site 40. Subchondral cyst and post-inflammatory osteophytes on the proximal aspect of *phalanx media* (2nd half 11th–1st half 14th c.).

**Figure 11 animals-14-00490-f011:**
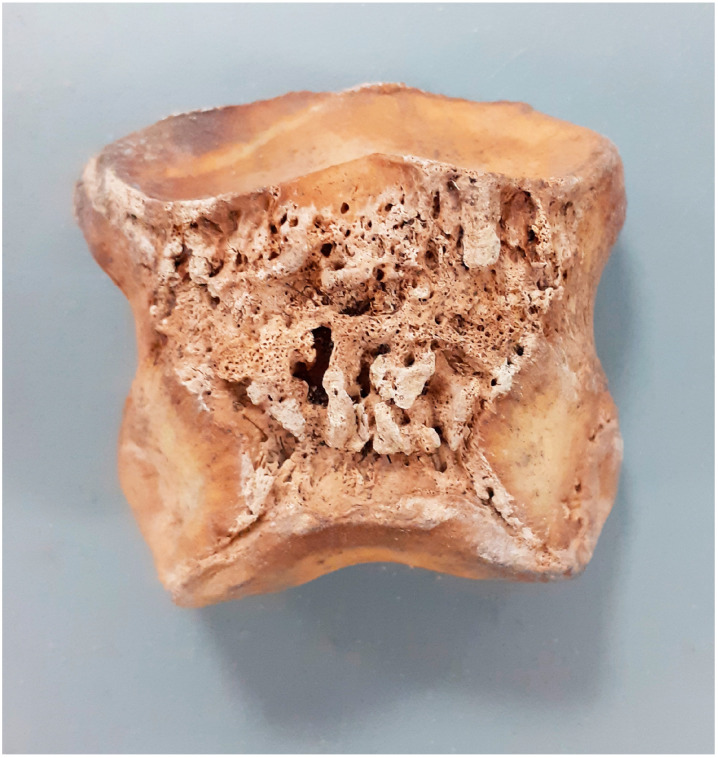
Massive osteophytes on the *phalanx media*.

**Figure 12 animals-14-00490-f012:**
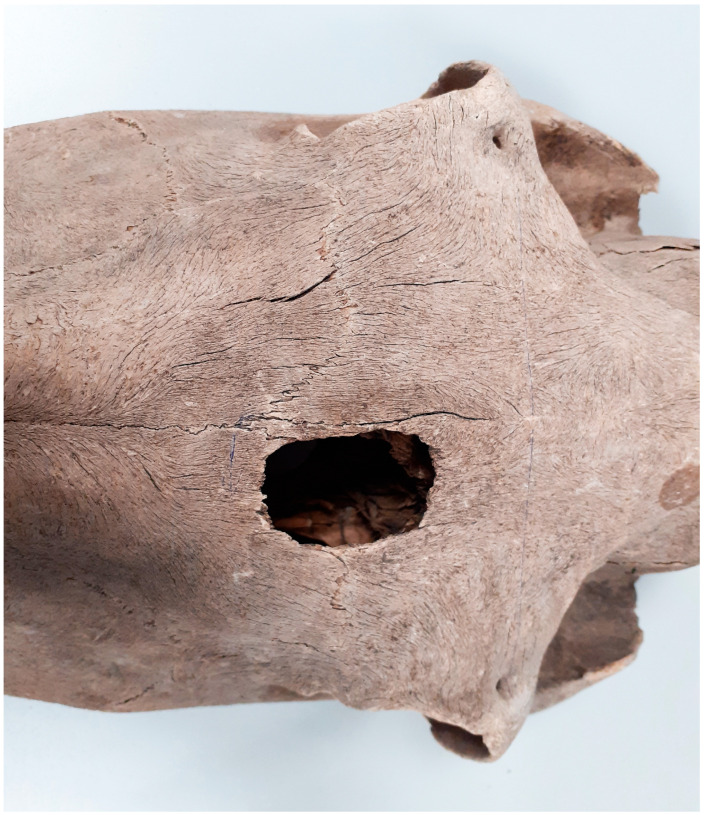
Ostrów Lednicki, site 2. Post-traumatic defect in the frontal bone (11th–mid 12th c.).

**Figure 13 animals-14-00490-f013:**
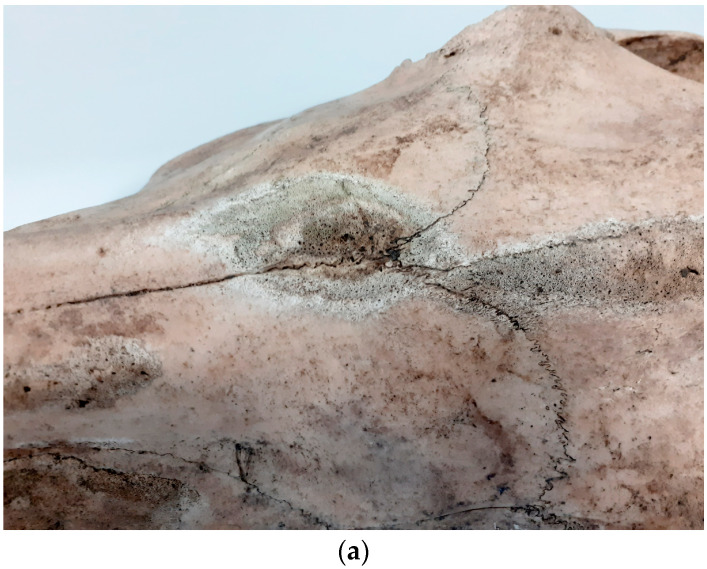
(**a**) Gdańsk, sit1 (Sukiennicza St.). Inflammatory changes with remodelling and bone loss on the maxillary and incisive bones (11th–mid 12th c.). (**b**) Inflammatory changes with remodelling and bone loss on the maxillary and incisive bones.

## Data Availability

Data are contained within the article and Appendix A.

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
