# Peer review of "Pathological Changes in Early Medieval Horses from Different Archaeological Sites in Poland"

_animals, 2024, doi:10.3390/ani14030490_

Round 1

Reviewer 1 Report

Comments and Suggestions for Authors

p 4 No. 121 (lig. longitudinale) ventrale) not  longitudilale

No. 129 which place of spondylosis (important)

Fig 5 possibility resign.

6 page No. 176 not tumor?  may be  tuber??

Fig 7 Advanced spavin (latin terminology of bones disease - chronica deformans tarsi) and place/location Daugnora, Thomas 2004 year).

No. 192 specinem (must be specimen)

page 8 No. 201 finger must be digital bones

page 9 No. 209 coronoid? bone in Nomina Anatomica Veterinaria not used us first termin. Digital bones  or Phalanx media, phalanx proximalis and phalanx distalis.

Page 8 Fig 9 (trigonum phalangis proximalis. Pastern bone (phalanx proximalis, proximal phalanx)

203 pastern bone  body of proximal phalanx

page 10 no 227 diameter of the hole?

Spatela ?? Patella

The third phalanx (old termin os  ungulare)

The second phalanx (os coronale)

Phalanx I (os compedale ) Ph I

IN FIG. must be  measure.

Fog. 11 No. 223 crown bone ?

Fig 13, 13a No. 249 incisive bone ( not incisor??? in text) it not dens incisivi.

No. 252 os incisivum (incisive bone)

Author Response

Dear Reviewer 1

Thank you for taking the time to read this manuscript and for your valuable comments.

p 4 No. 121 (lig. longitudinale) ventrale) not  longitudilale

Answer: Corrected as suggested by the Reviewer.

No. 129 which place of spondylosis (important)

Answer: spondylosis was observed on the dorsal surfaces of the thoracic and lumbar vertebrae.

Fig 5 possibility resign.

Answer: we decided to leave this figure.

6 page No. 176 not tumor?  may be  tuber??

Answer: Corrected as suggested by the Reviewer.

Fig 7 Advanced spavin (latin terminology of bones disease - chronica deformans tarsi) and place/location Daugnora, Thomas 2004 year).

Answer: term changed to bone spavin.

No. 192 specinem (must be specimen)

Answer: Corrected as suggested by the Reviewer.

page 8 No. 201 finger must be digital bones

Answer: we left this term unchanged.

page 9 No. 209 coronoid? bone in Nomina Anatomica Veterinaria not used us first termin. Digital bones  or Phalanx media, phalanx proximalis and phalanx distalis.

Answer: Corrected as suggested by the Reviewer.

Page 8 Fig 9 (trigonum phalangis proximalis. Pastern bone (phalanx proximalis, proximal phalanx)

Answer: Corrected as suggested by the Reviewer.

203 pastern bone  body of proximal phalanx

Answer: Corrected as suggested by the Reviewer.

page 10 no 227 diameter of the hole?

Answer: The dimensions of this hole are available in our article: Janeczek et al. 2023. Battle wound as a probable cause of the death of an early medieval horse in Ostrów Lednicki, Poland. International Journal of Paleopathology 40, 70–76.

Spatela ?? Patella

Answer: The correct term is bone spavin.We have corrected that in the manuscript.

The third phalanx (old termin os  ungulare)

Answer: Corrected as suggested by the Reviewer.

The second phalanx (os coronale)

Answer: Corrected as suggested by the Reviewer.

Phalanx I (os compedale ) Ph I

Answer: Corrected as suggested by the Reviewer.

IN FIG. must be  measure.

Answer: In our opinion, the figures may remain without measurements.

Fog. 11 No. 223 crown bone ?

Answer: We have corrected that in the manuscript.

Fig 13, 13a No. 249 incisive bone ( not incisor??? in text) it not dens incisivi.

Answer: Corrected as suggested by the Reviewer.

No. 252 os incisivum (incisive bone)

Answer: Corrected as suggested by the Reviewer.

Regards

Reviewer 2 Report

Comments and Suggestions for Authors

This is an easily readable and well-written article on equine pathologies. Horses have been very important animal to many societies, many researchers in various countries are are working on these "evergreen" animals. Thus, researches on horses, and especially on their pathologies, are always welcome. Animal health and pathologies are one of the most challenging areas of research, especially since many zooarchaeologists have no veterinary background, so this article will be useful for many specialists.
A few minor remarks: when a period is mentioned for the first time, the chronology could be specified.
It would be more convenient for the reader if the general statistics (%) of the various pathologies were placed in a table.
Thanks to the authors for an interesting and useful article, looking forward to new works and good luck!

Author Response

Dear Reviewer 2

Thank you for taking the time to read this manuscript and for your valuable comments.

This is an easily readable and well-written article on equine pathologies. Horses have been very important animal to many societies, many researchers in various countries are are working on these "evergreen" animals. Thus, researches on horses, and especially on their pathologies, are always welcome. Animal health and pathologies are one of the most challenging areas of research, especially since many zooarchaeologists have no veterinary background, so this article will be useful for many specialists.
A few minor remarks: when a period is mentioned for the first time, the chronology could be specified.

Answer: We have introduced chronology into the manuscript.

It would be more convenient for the reader if the general statistics (%) of the various pathologies were placed in a table.

Answer: In accordance with this suggestion, we have included the overall statistics (%) of various pathologies in the supplementary materials for this manuscript.

Regards

Reviewer 3 Report

Comments and Suggestions for Authors

The manuscript “Pathological Changes in Early Medieval Horses from Different 2 Archaeological Sites in Poland” serves as a fascinating compilation of equine pathologies, aligning with its title. The study involved the examination of material gathered from 20 archaeological sites. However, the authors omitted details regarding the overall number of remains discovered at each site and failed to specify how many of them exhibited pathologies. For instance, it remains unclear whether the 183 injuries identified correspond to an equal number of horses or a lesser quantity. Employing indices common in zooarchaeology, such as MNI and NISP, would be instrumental in addressing these uncertainties. Furthermore, the presentation of the ages of the studied animals is essential. It is assumed that the authors conducted a thorough examination of a significant sample to reveal the reported number of injuries, a noteworthy aspect that warrants recognition. Therefore, the numerical data must be included in the paper. Regrettably, the lack of a scale in the provided photographs impedes the capacity to compare findings with those from other archaeological sites. To address this issue, I recommend incorporating scales, possibly through the use of image data software, for instance. Nevertheless, the discussion is engaging and well-structured. The authors commendably elucidate the nature of the injuries and skillfully connect the data to activities that could have given rise to such injuries.

Author Response

Dear Reviewer 3

Thank you for taking the time to read this manuscript and for your valuable comments.

The manuscript “Pathological Changes in Early Medieval Horses from Different 2 Archaeological Sites in Poland” serves as a fascinating compilation of equine pathologies, aligning with its title. The study involved the examination of material gathered from 20 archaeological sites. However, the authors omitted details regarding the overall number of remains discovered at each site and failed to specify how many of them exhibited pathologies. For instance, it remains unclear whether the 183 injuries identified correspond to an equal number of horses or a lesser quantity. Employing indices common in zooarchaeology, such as MNI and NISP, would be instrumental in addressing these uncertainties. Furthermore, the presentation of the ages of the studied animals is essential. It is assumed that the authors conducted a thorough examination of a significant sample to reveal the reported number of injuries, a noteworthy aspect that warrants recognition. Therefore, the numerical data must be included in the paper. Regrettably, the lack of a scale in the provided photographs impedes the capacity to compare findings with those from other archaeological sites. To address this issue, I recommend incorporating scales, possibly through the use of image data software, for instance. Nevertheless, the discussion is engaging and well-structured. The authors commendably elucidate the nature of the injuries and skillfully connect the data to activities that could have given rise to such injuries.

Answer: We are grateful for your suggestions regarding our manuscript, thanks to which we could improve it significantly as well as correct our own inaccuracies. I think that both the form of the article and the information additionally included in the supplementary materials, will clarify any previous doubts.

Regards